# Obese Asthma Phenotype Is Associated with hsa-miR-26a-1-3p and hsa-miR-376a-3p Modulating the IGF Axis

**DOI:** 10.3390/ijms241411620

**Published:** 2023-07-18

**Authors:** Marta Gil-Martínez, Clara Lorente-Sorolla, José M. Rodrigo-Muñoz, Sara Naharro, Zahara García-de Castro, Joaquín Sastre, Marcela Valverde-Monge, Santiago Quirce, María L. Caballero, José M. Olaguibel, Victoria del Pozo

**Affiliations:** 1Immunoallergy Laboratory, Immunology Department, Instituto de Investigación Sanitaria Fundación Jiménez Díaz, Universidad Autónoma de Madrid (IIS-FJD, UAM), 28040 Madrid, Spain; marta.gilm@quironsalud.es (M.G.-M.); clara.lorente@quironsalud.es (C.L.-S.); jose.rodrigom@quironsalud.es (J.M.R.-M.); sara.naharrog@quironsalud.es (S.N.); zahara.garcia@quironsalud.es (Z.G.-d.C.); 2CIBER de Enfermedades Respiratorias (CIBERES), Instituto de Salud Carlos III (ISCIII), 28029 Madrid, Spain; jsastre@fjd.es (J.S.); marcela.valverde@quironsalud.es (M.V.-M.); squirce@gmail.com (S.Q.); mlcsoto@hotmail.com (M.L.C.); jmolaguibel@telefonica.net (J.M.O.); 3Allergy Department, Hospital Universitario Fundación Jiménez Díaz, 28040 Madrid, Spain; 4Department of Allergy, Hospital Universitario La Paz, IdiPAZ, 28046 Madrid, Spain; 5Severe Asthma Unit, Department of Allergy, Hospital Universitario de Navarra, NavarraBiomed, 31008 Pamplona, Spain; 6Department of Medicine, Faculty of Medicine, Universidad Autónoma de Madrid (UAM), 28049 Madrid, Spain

**Keywords:** asthma phenotypes/endotypes, eosinophilic asthma, obese asthma, serum microRNAs, biomarkers, IGF axis

## Abstract

Clarifying inflammatory processes and categorising asthma into phenotypes and endotypes improves asthma management. Obesity worsens severe asthma and reduces quality of life, although its specific molecular impact remains unclear. We previously demonstrated that hsa-miR-26a-1-3p and hsa-miR-376a-3p, biomarkers related to an inflammatory profile, discriminate eosinophilic from non-eosinophilic asthmatics. We aimed to study hsa-miR-26a-1-3p, hsa-miR-376a-3p, and their target genes in asthmatic subjects with or without obesity to find biomarkers and comprehend obese asthma mechanisms. Lung tissue samples were obtained from asthmatic patients (n = 16) and healthy subjects (n = 20). We measured miRNA expression using RT-qPCR and protein levels (IGF axis) by ELISA in confirmation samples from eosinophilic (n = 38) and non-eosinophilic (n = 39) obese (n = 26) and non-obese (n = 51) asthma patients. Asthmatic lungs showed higher hsa-miR-26a-1-3p and hsa-miR-376a-3p expression than healthy lungs. A study of seven genes regulated by these miRNAs revealed differential expression of *IGFBP3* between asthma patients and healthy individuals. In obese asthma patients, we found higher hsa-miR-26a-1-3p and IGF-1R values and lower values for hsa-miR-376a-3p and IGFBP-3. Hsa-miR-26a-1-3p and IGFBP-3 were directly and inversely correlated with body mass index, respectively. Hsa-miR-26a-1-3p and hsa-miR-376a-3p could be used as biomarkers to phenotype patients with eosinophilic and non-eosinophilic asthma in relation to comorbid obesity.

## 1. Introduction

Asthma is a heterogeneous disease of the airways characterised by chronic airway inflammation leading to respiratory symptoms [1]. Asthma comprises an array of phenotypes driven by different mechanistic pathways, known as endotypes [2]. The asthma subtype most commonly associated with the T2 inflammatory profile (T2-high asthma) is severe eosinophilic asthma, in which patients experience recurrent exacerbations despite high doses of inhaled corticosteroids or even occasional oral corticosteroids [3].

Asthma is frequently accompanied by a variety of comorbidities that influence the severity of the disease. These are associated with a clinically significant reduction in quality of life (QoL) and exacerbate the rate of morbidity and the economic burden of the disease [4]. These comorbidities have been categorised based on the involvement of respiratory and/or extra-respiratory domains [5]. Obesity is a major public health concern as well as an extra-respiratory risk factor for both asthma incidence and prevalence. The relationship between obesity and asthma is the result of a complex interplay between biological, physiological, and environmental factors [6]. Nowadays, with the continuous rise in obesity, particularly among children, the obese asthma phenotype is attracting great interest [7]. Both T2 and non-T2 inflammation can be encountered in asthma patients with obesity [8]. Obese asthmatic patients are primarily associated with a non-eosinophilic phenotype and are characterised by a state of low-grade systemic inflammation marked by the activation of M1 macrophages and CD8^+^ T cells and an increase in inflammatory indicators such as Toll-like receptor 4 (TLR4), interleukin (IL)-1b, IL-6 and IL-17, interferon (IFN)-γ, tumour necrosis factor (TNF)-α, leptin, and resistin. Asthma, which is often severe in patients who are obese, may be somewhat resistant to current treatment options, including corticosteroids, thus limiting possible therapeutic approaches and complicating efforts to control the disease [8,9,10].

It is unclear which pathophysiological mechanisms underlie the development of obesity as a comorbidity and enhancer of asthma severity and how these mechanisms are related, and further research on the endophenotype of obese patients with severe eosinophilic asthma is needed [10]. Biomarkers are highly useful for phenotyping and endotyping asthma, as they allow for personalised therapies to be offered to patients [11]. Currently, there is increasing interest in using microRNA (miRNA) profiles as biomarkers of disease [12]. miRNAs, which are small, non-coding RNA molecules, are engaged in regulating gene expression by inhibiting protein translation [13]. Previous studies have demonstrated that miRNAs hold great potential as asthma mediators and biomarkers [14]. In previous research, we used serum samples from patients with eosinophilic (n = 40) and non-eosinophilic (n = 36) asthma to evaluate miRNA expression by next-generation sequencing (miRNA-seq), with subsequent validation by RT-qPCR. After these analyses, we demonstrated that hsa-miR-26a-1-3p and hsa-miR-376a-3p were able to discriminate between patients with eosinophilic and non-eosinophilic asthma [11].

The aim of this study was to analyse the expression of these miRNAs and their targets in subjects with eosinophilic vs. non-eosinophilic asthma, with or without comorbid obesity, in order to determine whether these miRNAs are associated with differential inflammatory pathways of asthma.

## 2. Results

### 2.1. Characteristics of the Study Population

Descriptive data on the demographic, inflammatory, functional, and clinical characteristics of all individuals studied are summarised in Appendix A (population 1) and Appendix A (population 2).

Contrasting the group of asthmatic patients and control individuals based on lung biopsy samples (Appendix A), age was the only significant difference observed (52.4 vs. 63.7 years, *p* < 0.05).

As evidenced in Appendix A, with respect to functional and clinical parameters, the serum samples of obese and non-obese asthmatic patients showed significant differences in FEV1/FVC (%) (86.5 vs. 78.8, *p* < 0.05) and in severe and moderate disease severity (32% vs. 56.8%, and 44% vs. 20.5%; *p* < 0.05). The functional and clinical parameters of these patients according to the four study groups are shown in Table 1 (population 2, obese and non-obese, further subdivided into eosinophilic and non-eosinophilic), showing significant differences in FeNO levels (*p* < 0.0001) and in the rates of severe asthma (*p* < 0.05). None of the other demographic, inflammatory, functional, or clinical characteristics presented statistically significant differences (Appendix A and Table 1).

### 2.2. miRNA and Gene Expression in Lung Tissue Samples

We found statistically significant differences in the expression of hsa-miR-26a-1-3p and hsa-miR-376a-3p in lung biopsies between asthmatic patients and healthy subjects (*p* < 0.01 and *p* < 0.001, respectively), with higher expressions observed in asthmatic patients (Figure 1a). This confirms that serum hsa-miR-26a-1-3p and hsa-miR-376a-3p are linked to asthma, suggesting that these miRNAs could possibly serve as biomarkers of the disease.

We subsequently conducted an in silico analysis using DIANA-miRPath v3.0 to determine the implication of these two differentially expressed miRNAs in biological processes and to identify potential target genes. We found that hsa-miR-26a-1-3p and hsa-miR-376a-3p control several signalling pathways: prion diseases (*p* < 0.0001), p53 signalling pathway and non-small cell lung cancer (*p* < 0.05), regulated by hsa-miR-26a-1-3p, and extracellular matrix (ECM)-receptor interaction (*p* < 0.0001), influenced by hsa-miR-376a-3p. Of these, the p53 signalling pathway and extracellular matrix (ECM)-receptor interaction are critical pathways related to asthma pathogenesis. Since *CDK6*, *CCND1*, and *IGFBP3*, involved in the p53 signalling pathway, are target genes of hsa-miR-26a-1-3p, and *COL3A1*, *COL6A2*, *COL1A1*, and *COL6A3*, implicated in the extracellular matrix (ECM)-receptor interaction signalling pathway, are targeted by hsa-miR-376a-3p, RT-qPCR was performed on lung biopsy samples from asthmatic patients and healthy individuals. The results of this analysis revealed that only *IGFBP3* was significantly increased in asthmatic individuals in comparison to healthy subjects (*p* < 0.01; Figure 1b).

To establish the association between miRNA molecules and their target gene expression levels (ΔCt), a lung-tissue correlation analysis was carried out. We verified a significant, direct correlation for the total population between hsa-miR-26a-1-3p and its target gene, *IGFBP3* (Pearson r = 0.36, *p* < 0.05), instead of detecting, as usual, an inverse miRNA-target gene correlation. Since miRNAs generally behave as silencers of their target mRNAs, a possible explanation would be that miRNA–miRNA relationships are complex, and the lung tissue samples in which they were studied differ from each other (influence of genetic and/or environmental factors). There may be other miRNAs involved in the modulation of a particular mRNA (IGFBP-3 mRNA) and there could also be an interaction of other target genes of this miRNA (hsa-miR-26a-1-3p) that could be inhibitors of *IGFBP3*, in this case, highlighting the intricacy of post-transcriptional miRNA–mRNA regulation throughout the whole tissue.

### 2.3. miRNA Expression and Extracellular Protein Concentration in Serum Samples

To further investigate and confirm the relationship between these miRNAs and the *IGFBP3* gene, we analysed the serum expression of two soluble extracellular proteins, IGFBP-3 (encoded by *IGFBP3*, the target gene of hsa-miR-26a-1-3p) and IGF-1R (encoded by *IGF1R*, targeted by hsa-miR-376a-3p), both of which are implicated, in the IGF axis, in patients with eosinophilic and non-eosinophilic asthma. No significant differences in target protein levels were detected between the eosinophilic and non-eosinophilic groups (data not shown), although statistically significant differences were observed in IGFBP-3 (*p* < 0.01) and IGF-1R (*p* < 0.05) levels among obese and non-obese asthmatic patients: IGFBP-3 showed higher levels in non-obese subjects, while IGF-1R levels were greater in obese individuals (Figure 2a,b). Regarding the levels of these proteins when subdividing patients with and without eosinophilic asthma into obese and non-obese subjects, IGFBP-3 was more concentrated in non-obese individuals than in obese subjects (*p* < 0.05) among patients with eosinophilic asthma and those with non-eosinophilic asthma (Figure 2a); on the other hand, IGF-1R was higher in obese individuals than in non-obese subjects (*p* = 0.05 and *p* < 0.05, in that order), and obese patients with eosinophilic asthma had higher amounts of IGF-1R than the rest of the groups (Figure 2b).

Next, we assessed whether hsa-miR-26a-1-3p and hsa-miR-376a-3p are modulated by obesity status in asthmatic patients with and without eosinophilia. A non-significant increase in hsa-miR-26a-1-3p expression and lower expression of hsa-miR-376a-3p (*p* = 0.07) was found in this group of obese subjects (Figure 2c,d). When we subdivided patients with eosinophilic and non-eosinophilic asthma into obese and non-obese individuals, we observed that both miRNAs were more highly expressed in patients with eosinophilic asthma than in those with non-eosinophilic asthma (*p* < 0.05 and *p* < 0.001, and *p* < 0.05 and *p* < 0.01, respectively; Figure 2c,d). Specifically, hsa-miR-26a-1-3p exhibited higher expression in obese subjects than in non-obese individuals (*p* < 0.01) and in patients with non-eosinophilic asthma (Figure 2c), while hsa-miR-376a-3p levels were highest among non-obese eosinophilic asthmatics.

### 2.4. Correlation of miRNAs and Soluble Extracellular Proteins with Clinical Parameters

Because the measured soluble extracellular proteins are engaged in the IGF axis, which is metabolism-related, we analysed our data to determine if BMI correlated with serum levels of these proteins and miRNAs (Figure 3). A negative correlation was identified between IGFBP-3 and BMI in the group of patients with eosinophilic asthma (Pearson r = −0.43, *p* < 0.01; Figure 3a) and in the total study population (Pearson r = −0.34, *p* < 0.01; Figure 3c); while hsa-miR-26a-1-3p correlated directly with BMI in patients with non-eosinophilic disease (Pearson r = 0.86, *p* < 0.0001; Figure 3b) and in the total population (Pearson r = 0.38, *p* < 0.05; Figure 3c).

### 2.5. ROC Curves to Establish Potential Biomarkers of Inflammatory Endotype in Asthma

ROC curve analyses for the two miRNAs (Table 2) were performed for two-by-two group comparisons (obese and non-obese eosinophilic asthma patients, obese and non-obese non-eosinophilic asthma patients, and the entire population of obese and non-obese asthma patients), and the AUC were estimated.

Hsa-miR-26a-1-3p presented an AUC value of 0.91 for the comparison between non-eosinophilic and eosinophilic asthma within the non-obese group (*p* < 0.001). An AUC of 1.00 was found for the comparisons between obese and non-obese, non-eosinophilic asthmatics (*p* < 0.01) and between the non-obese, non-eosinophilic and obese eosinophilic asthma patients (*p* < 0.05). Hsa-miR-376a-3p showed AUC values of 0.71 and 0.75, respectively, for the comparisons between non-eosinophilic and eosinophilic subjects in the non-obese asthma group (*p* < 0.05) and between obese non-eosinophilic and non-obese eosinophilic asthmatic individuals (*p* < 0.01).

## 3. Discussion

The results of this study reveal that hsa-miR-26a-1-3p and hsa-miR-376a-3p are post-transcriptional modulators of the IGF axis and potential epigenetic biomarkers to differentiate asthma patients by inflammatory condition (eosinophilia and/or obesity).

Severe asthma is an inherently heterogeneous disease that encompasses several clinical phenotypes that may have several shared pathophysiological mechanisms, known as endotypes of asthma [15]. Therefore, inflammatory endotype characterisation should be a crucial component of the algorithm used to assess and manage severe disease [16]. The discovery and use of biomarkers to define phenotypes and endotypes of severe asthma and guide therapeutic strategy is increasingly necessary for clinicians, as an individual approach to disease management and personalised medicine is essential [17]. Eosinophilic phenotypes and T2-associated endotypes with eosinophilic inflammation are currently identified in clinical practice by biomarkers, including peripheral blood and sputum eosinophil count, total serum IgE levels, FeNO value, periostin, and miRNAs [18]. We observed that serum samples from both obese and non-obese patients with eosinophilic asthma showed markedly higher FeNO scores compared to non-eosinophilic asthmatic patients. As previous research has found a relationship between FeNO and the eosinophilic phenotype in asthma, FeNO, in combination with blood eosinophils, can accurately predict eosinophilic airway inflammation [19,20]. Also, as demonstrated by our results (non-obese, non-eosinophilic asthma group), elevated FeNO values and increased blood eosinophil counts are associated with increased severity of asthmatic disease, worse lung function, and an increased risk of asthma exacerbation [21].

It has been established that severe asthma is a complex clinical condition not exclusively related to airway inflammation and response to treatment [22]. Asthma is often associated with various comorbidities which can worsen asthma symptoms and affect asthma severity [23]. For patients with asthma, comorbid conditions lead to an enhanced disease burden and, consequently, a reduced QoL [24]. The involvement of other disorders is likely to lead to polypharmacy, which can negatively impact treatment adherence and asthma control [25,26] Additionally, comorbid conditions complicate the diagnosis and management of asthmatic patients and carry a considerable economic cost [23]. Obesity is a common comorbidity in children and adults with asthma, and both obesity and asthma have risen dramatically in prevalence [27,28]. Obesity is defined as having a BMI of 30 kg/m^2^ or greater [29]. Obese asthmatics are generally characterised by more severe asthma, are less responsive to treatment, and have poorer asthma control, according to the asthma control test (ACT) questionnaires [30], although this does not appear to be a determining factor in the risk of asthma exacerbations with regard to disease severity [31]. This absence of a link between obesity status and exacerbations appears to be in line with our results, as both obese asthmatic patient groups, compared to the two non-obese asthmatic patient groups, had similar or higher asthma severity (80% and 73.3%, respectively; 88.5% and 61.1% of patients with severe and moderate asthma), lower ACT scores (20.5 and 21, 21 and 22), and a lower, or roughly similar, presence of exacerbations (50% and 50%, 60.7% and 39.1%). As cited earlier, our findings indicate more frequent occurrences of exacerbations among obese women (19/26) and late-onset asthmatics (17/24) [12]. Also, since no significant differences are observed in relation to patient therapy, it could be considered that the changes in miRNA expression are not due to the treatment. However, it is worth mentioning that, surprisingly, in contrast to what has been described in the literature in relation to lung function, that obese asthmatic patients have worse lung function than non-obese asthmatic patients. In our population the opposite is shown, obese asthma patients show more improved lung function than non-obese asthma subjects [32], which may be due, to some extent, to the fact that there are more non-obese than obese asthmatic patients and, within the group of non-obese asthmatic patients there was a high number of patients with eosinophilic asthma, which may cause these non-obese asthmatic patients to have lower pulmonary function since, subdividing obese and non-obese asthmatic patients into eosinophilic and non-eosinophilic categories, the eosinophilic asthmatic patient groups showed poorer lung function than the non-eosinophilic asthmatic patient groups, contributing to a decrease in the overall group’s lung function. Moreover, according to the percentage of neutrophils in sputum in our study, the 27% of patients with obese asthma are individuals with a neutrophilic component. The differences in clinical and inflammatory findings between the four patient groups may be due to the combined influence of the different inflammation profiles in obese asthma [12] and the pathophysiological differences between eosinophilic and non-eosinophilic asthma [33].

Biomarkers are promising tools for categorising asthma patients into different phenotypes and endotypes, as they enable the identification of potential therapeutic targets and make it possible to deliver personalised therapy [17]; of these tools, miRNAs could be an effective biomarker [34]. In this study we found expression differences of the two miRNAs, hsa-miR-26a-1-3p and hsa-miR-376a-3p, in lung tissue samples from asthmatic individuals and healthy subjects. The same difference was previously observed in serum from patients with eosinophilic asthma and others with non-eosinophilic asthma [11]. This finding supports the involvement of these miRNAs in asthma, as they were detected in target tissue, and further indicates that the two miRNAs could serve as potential biomarkers to classify patients into either group. In terms of downstream effects, in silico experiments revealed that hsa-miR-26a-1-3p and hsa-miR-376a-3p significantly modulate two pathways associated with asthma pathogenesis that play a key role in the development of airway inflammation and asthma remodelling [35,36]. Also, of the target genes selected for study due to their relationship with differential expression of these two miRNAs, only *IGFBP3* displayed differential expression between lung biopsy samples from asthmatic individuals and healthy subjects, with higher values found in asthmatic individuals; the expression levels of the miRNA and its target gene were correlated, emphasising the involvement of this inflammation-related gene in asthma pathology, as has been previously described [37]. Moreover, we found differences in the expression of two soluble extracellular proteins, IGFBP-3 and IGF-1R, which are encoded by target genes of these miRNAs, in the serum of subjects with eosinophilic asthma and in individuals with non-eosinophilic asthma, further separated into obese and non-obese asthmatic patients.

Growth hormone (GH), which is deficient in obese patients, stimulates the secretion of insulin-like growth factor I (IGF-1) in most tissues. Jointly, GH and IGF-1 exert strong actions on fat, protein, and glucose metabolism [38]. The actions of IGF-1 are typically mediated through interactions with its receptor (IGF-1R), resulting in the activation of intracellular signalling pathways, including MAP kinase and PI3 kinase/AKT [39]. In addition, the bioavailability, half-life, and actions of IGF-1 in circulation are regulated through its binding to one of the members of the IGF binding proteins, of which IGFBP-3 is the most abundant in plasma and the one with the highest affinity for IGF-1 [40]. Alterations in the IGF axis may be implicated in a variety of pathological conditions, including obesity [41]. Furthermore, a link between asthma and obesity has been recognised alongside other markers of the metabolic syndrome, such as insulin resistance and hypertension [42]. It has also been proposed that insulin resistance may have an effect on asthma risk [43]. Peters and co-workers observed that, among asthma patients participating in a severe asthma research programme, patients with insulin resistance had poorer asthma-control test scores and greater deterioration in lung-function test responses [44]. Finally, it is well known that both insulin and IGF-1 can mutually activate insulin and IGF-1R receptors, sharing downstream signalling pathways but with different biological effects [45].

Regarding our study on obese asthma molecular mechanisms, on the one hand, hsa-miR-26a-1-3p showed higher expression in subjects with eosinophilic asthma and in the subgroups of obese individuals, while the concentration of the soluble extracellular protein IGFBP-3, which is encoded by the target gene of this miRNA, as expected in terms of miRNA-gene interaction, was lower in the non-obese patient groups. Also, since IGFBP-3 is a molecule that displays anti-inflammatory action [46], a greater concentration was expected in subjects with no additional inflammatory condition such as obesity. As reported in other studies [47], hsa-miR-376a-3p showed greater expression in non-obese subjects and in asthmatic patients without additional comorbid conditions, with expression increasing in eosinophilic asthmatics. As in the previous case, there is an inverse miRNA-gene relation, such that the concentration of IGF-1R, which is encoded by the target gene of hsa-miR-376a-3p, was elevated in obese individuals, and specifically in obese patients with eosinophilic asthma. This result supports previous reports indicating that the higher expression of this molecule results in more prominent inflammatory responses [48].

Since IGFBP-3 and IGF-1R are involved in the IGF axis, which is closely related to metabolism, we studied the correlation of the expression of hsa-miR-26a-1-3p and hsa-miR-376a-3p, in addition to these two soluble extracellular proteins encoded by their respective target genes, and established their correlation with clinical parameters. The results suggested that these molecules could be relevant in phenotypically distinguishing patients based on the inflammatory component and thus be used in asthma diagnosis.

Lastly, based on the AUC values calculated, hsa-miR-26a-1-3p could be a highly valuable biomarker in differentiating obese from non-obese subjects among patients with non-eosinophilic asthma or in discriminating individuals with no inflammatory condition (non-obese, non-eosinophilic asthma subjects) from individuals with combined inflammatory states (obese eosinophilic asthma subjects). Hsa-miR-26a-1-3p is also effective in distinguishing non-eosinophilic from eosinophilic individuals within the group of non-obese asthma patients. Therefore, this miRNA can discern the inflammatory conditions of obesity or eosinophilia (individually) and can distinguish the presence of both inflammatory processes from cases with no inflammatory condition. Based on the AUC score found, hsa-miR-376a-3p could be an acceptable biomarker to differentiate non-eosinophilic from eosinophilic subjects within the group of non-obese asthma patients (i.e., eosinophilia in the absence of obesity) or to separate individuals with one inflammatory state (i.e., non-obese, non-eosinophilic patients) from subjects with the other (non-obese eosinophilic patients).

We wish to point out that a limitation of this study is the low number of samples from obese individuals, mainly patients with eosinophilic asthma, compared to the number of samples from non-obese subjects (patients with eosinophilic asthma and non-eosinophilic asthma). A more even sample size in the four groups could provide more consistent results. Another limitation of our study is the existence of significant differences in age between the asthmatic and control groups of subjects, whose lung tissue samples were included for analysis, as this was not evaluated as a confounding factor. However, although there were significant age differences between these two groups, in the obese and non-obese groups (with a larger number of subjects), from which serum samples were included, or in the four groups after being subdivided, no significant age differences were observed; therefore, it is possible that age does not significantly affect miRNA expression. In addition, unfortunately, as the samples were from patients belonging to biobanks, we did not have all the clinical data available.

## 4. Materials and Methods

### 4.1. Study Subjects and Sample Collection

We obtained lung biopsy samples from 16 asthmatic patients and 20 control subject donors (population 1) to examine miRNAs and gene expression. Biopsy samples were provided by the CIBERES Pulmonary Biobank Consortium (PT13/0010/0030), a network currently made up of 12 tertiary Spanish hospitals (www.ciberes.org; accessed on 5 June 2023) listed in the Acknowledgements section and integrated in the Spanish National Biobanks Network. Lung biopsies were processed in accordance with standard operating procedures, and subsequent approval was granted for processing by the local ethics and scientific committees (ref. B.0000471 Registro Nacional de Biobancos—ISCIII; CIBERES Pulmonary Biobank Consortium started its activity in 2008 and has been ISO9001:2015 (formerly ISO9001:2008) certified since 2012.).

Lung tissue samples were preserved in RNAlater stabilisation solution at −80 °C until use.

A cohort of 77 subjects with diagnosed asthma (population 2) was selected for confirmation studies. This cohort was recruited from the allergy and pulmonology units of Fundación Jiménez Díaz Hospital in Madrid (Spain); 10 of these individuals were phenotyped as obese patients with eosinophilic asthma, 28 as non-obese patients with eosinophilic asthma, 16 as obese patients with non-eosinophilic asthma, and 23 as non-obese patients with non-eosinophilic asthma. Descriptive data representing the demographic, inflammatory, functional, and clinical characteristics of study subjects were collected.

The following scheme (Appendix A) shows the groups of the two studies (with the exact number of patients per group) and the material (lung tissue or serum) used from each patient in each phase of the study, facilitating the understanding of the design.

Serum samples were obtained by blood clotting in plain, additive-free tubes which were subsequently centrifuged at 3000 rpm for 10 min at 4 °C and stored at −80 °C until use.

The inclusion criteria were as follows: (i) acceptance to participate, providing signed informed consent; (ii) asthma diagnosis in accordance with the 2021 GINA criteria [49]; and (iii) age between 18 and 75 years. Eosinophilic asthma was determined by counts of ≥500 eosinophils/mm^3^ in peripheral blood, while non-eosinophilic patients had ≤200 eosinophils/mm^3^. Participants were considered obese if they had a body mass index (BMI) ≥30 kg/m^2^, and non-obese if BMI was <30 kg/m^2^.

The study was conducted in accordance with the tenets of the Declaration of Helsinki, and the protocol was approved by the CEIm of the participating hospital ethics committees (EOH2014/48, 4 December 2014).

### 4.2. miRNA and mRNA Analysis by RT-qPCR

Four µL of serum miRNA samples or 30 ng of miRNA-enriched lung tissue RNA were reverse-transcribed to cDNA using the miRCURY LNA RT Kit (Qiagen, Hilden, Germany) following the manufacturer’s protocol. Synthetic Uni-Sp6 and miRNA cel-miR-39-3p (Qiagen, Hilden, Germany) were used to verify correct reverse transcription. The cDNA obtained was stored at −20 °C until use. Then, miRNA expression was evaluated by qPCR using the miRCURY LNA SYBR Green PCR Kit (Qiagen) as indicated in the instructions. For this purpose, we used 3 μL of cDNA from the serum or lung tissue miRNAs diluted in 1:30 or 1:60 in RNase-free water, respectively, according to the producer’s recommendations. To ensure that differences in concentration, due to choosing the same volume in the case of serum samples, did not affect the miRNA expression measurements, the samples were normalised with endogenous control genes, which functioned as a control for the qPCR reaction. The following probes (Qiagen) were used for miRNA expression analysis in the serum and lung tissue: hsa-miR-26a-1-3p and hsa-miR-376a-3p. Additionally, hsa-miR-103a-3p, hsa-miR-191-5p, SP6, cel-miR-39-3p, and U6 (Qiagen) were selected as housekeeping miRNAs. The program GraphPad Prism^®^ v6-8.0 was used to perform a homogeneity test of Ct means of the samples from each of the comparison groups to select endogenous and/or exogenous genes as internal control of the qPCR reaction. In addition, a normalisation analysis was carried out with the use of the Bestkeeper tool. All samples were run in triplicate, and reactions were performed in a LightCycler^®^ 96 thermal cycler (Roche, Basel, Switzerland). Cycle threshold (Ct) values were analysed with LightCycler^®^ 96 SW 1.1 (Roche) software, and miRNA expression was calculated by normalising levels of expression to the endogenous miRNA controls. Relative quantification of differences in expression (RQ = 2^−ΔΔCt^; where ΔΔCt = ΔCt_eosinophilic asthmatics_ − ΔCt_non-eosinophilic asthmatics_ and ΔCt = Ct_miRNA_ − Ct_housekeeping miRNAs_) was carried out by means of the ΔΔCt method [50].

For gene expression analysis in lung tissue, 500 ng of RNA quantified by Nanodrop^®^ ND-1000 spectrophotometer (Thermo Fisher Scientific, Waltham, MA, USA) was reverse transcribed using the High-Capacity cDNA Reverse Transcription Kit (Applied Biosystems, Foster City, CA, USA), followed by qPCR analysis according to manufacturer guidelines on a StepOne™ Real-Time PCR System (Applied Biosystems). TaqMan™ gene expression probes were purchased for *CDK6*, *CCND1*, *IGFBP3*, *COL3A1*, *COL6A2*, *COL1A1*, *COL3A3*, and *GAPDH* using TaqMan™ Gene Expression Master Mix (Applied Biosystems). Gene expression was calculated by normalising to the endogenous control gene *GAPDH* applying the 2^−ΔΔCt^ method as reported previously (RQ = 2^−ΔΔCt^; where ΔΔCt = ΔCt_asthmatics_ − ΔCt_healthy subjects_) [50].

### 4.3. Pathway Enrichment Analyses

In order to identify the target genes related to asthma disease, hsa-miR-26-a-1-3p and hsa-miR-376a-3p, and differentially expressed miRNAs, a pathway enrichment analysis of the dysregulated miRNAs was performed using the DIANA-miRPath v3.0 online bioinformatic resource [51]. Relevant pathways for asthma pathology were represented when a *p*-value < 0.05, and the involved genes were analysed by RT-PCR, as previously mentioned.

### 4.4. Protein Detection by ELISA

Zehavi et al., identified the *IGF1R* gene as a target of hsa-miR-376a and hsa-miR-376c miRNAs. Since the protein encoded by the *IGF1R* gene is involved in the IGF axis, it was chosen, together with IGFBP-3, for ELISA evaluation [52].

For a determination of the soluble extracellular proteins IGFBP-3 and IGF-1R (genes encoded which are targeted by both hsa-miR-26a-1-3p and hsa-miR-376a-3p), 10 µL (dilution 1:100) or 100 µL of serum samples, in that order, were used for ELISA, performed with the Human IGFBP-3 Quantikine ELISA kit (R&D Systems, Minneapolis, MN, USA) and Human IGF-I R/IGF-1R DuoSet ELISA Kit (Bio-Techne, Minneapolis, MN, USA), respectively, following the manufacturers’ instructions.

The optical density of each well was determined using a Tecan Infinite^®^ F200 microplate reader (TECAN, Männedorf, Switzerland) adjusted to 450 nm, with the wavelength correction set to 570 nm.

### 4.5. Statistical Analysis

Statistical analyses and graphs were created with GraphPad Prism^®^ v6-8.0 (GraphPad Software Inc., San Diego, CA, USA).

The results are shown as mean (standard deviation, SD) or median (interquartile range, IQR) values. For all statistical analyses, differences showing *p* < 0.05 were considered significant. Normality was analysed by means of the Shapiro–Wilk test. For continuous variables, comparisons of normally distributed data among non-paired groups were performed via an unpaired t test (equal to SD) and t test with Welch’s correction (different SD), and non-normally distributed data and non-paired groups were compared by Mann–Whitney test.

Multiple comparisons between groups with data following a normal distribution were carried out with one-way ANOVA test (equal to SD) and Brown–Forsythe and Welch ANOVA test (different SD) without Welch’s correction test or with Kruskal–Wallis test without Dunn’s correction for non-normally distributed data.

Additionally, correlations among miRNA expression levels (ΔCt), soluble extracellular protein concentration, and clinical parameters (quantitative variables) were estimated by Spearman (non-normally distributed data) or Pearson (normally distributed data) correlations; one-way ANOVA and Chi-square test (contingency table) were applied to test the null hypothesis of the independence of groups and some demographic, inflammatory, functional, and clinical (quantitative and qualitative) variables. Finally, the expression profile (ΔCt) of each differentially expressed miRNA was used to create receiver operator characteristic (ROC) curves to evaluate miRNA performance as biomarkers; an area under the curve (AUC) of 0.7 indicated an acceptable biomarker.

## 5. Conclusions

In summary, this is the first study to describe differences in the expression of the previously validated serum miRNAs, hsa-miR-26a-1-3p and hsa-miR-376a-3p, in lung tissue samples. Findings from this study support their potential use as biomarkers to distinguish between obese and non-obese non-eosinophilic asthma patients, and in non-obese asthma patients, and to catalogue eosinophilic and non-eosinophilic patients. Finally, the findings can be used to discriminate non-eosinophilic asthma patients without obesity and eosinophilic asthma patients with obesity.

## Figures and Tables

**Figure 1 ijms-24-11620-f001:**
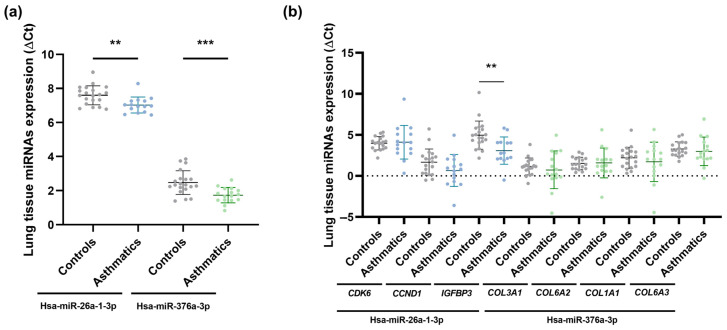
Lung biopsy sample miRNA expression (ΔCt) in asthmatic and non-asthmatic (controls) subjects. Hsa-miR-26a-1-3p and hsa-miR-376a-3p were differentially expressed between the two groups, showing higher expression in asthmatic individuals (**a**); and gene expression (ΔCt) measured in lung tissue samples from asthmatic and non-asthmatic (controls) subjects. Only *IGFBP3* exhibited differential expression between the two groups, being overexpressed in the asthma group (**b**). **, *p* < 0.01; ***, *p* < 0.001.

**Figure 2 ijms-24-11620-f002:**
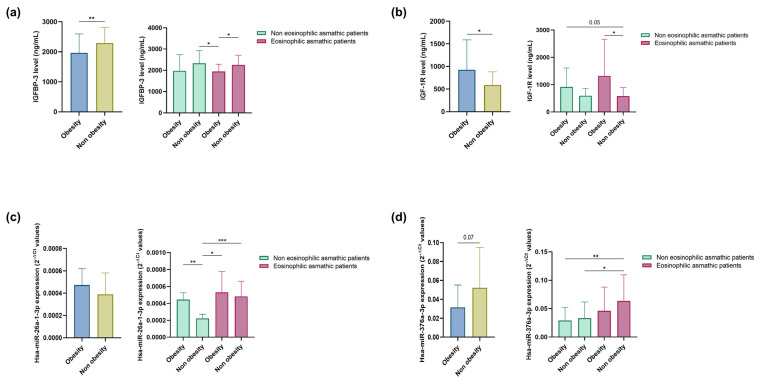
Serum IGFBP-3 (**a**); and IGF-1R (**b**) concentration. Serum hsa-miR-26a-1-3p (**c**); and hsa-miR-376a-3p (**d**) (2^−ΔCt^). Differential expression of miRNAs and soluble extracellular proteins was observed between obese and non-obese eosinophilic asthma patients and obese and non-obese non-eosinophilic asthma patients. *, *p* < 0.05; **, *p* < 0.01; ***, *p* < 0.001.

**Figure 3 ijms-24-11620-f003:**
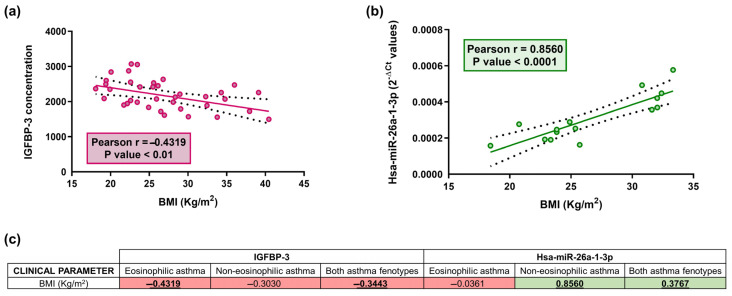
Relative IGFBP-3 concentration correlates inversely with BMI in serum samples (*p* < 0.01) in eosinophilic asthmatic patients (**a**); a direct correlation between relative hsa-miR-26a-1-3p expression (2^−ΔCt^) and BMI was reported in serum samples (*p* < 0.0001) in non-eosinophilic asthmatic patients (**b**); Dots (red and green) are individual values, black dashed line indicates error bars and red/green solid line shows linear regression (**a**,**b**). Table showing the correlation coefficient (r) between BMI parameter and serum IGBP-3 concentration and hsa-miR-26a-1-3p expression levels (2^−ΔCt^) (**c**). The red color indicates negative correlations, whereas green shows positive correlations (**c**). Bold and underlined values are statistically significant (*p* < 0.05).

**Table 1 ijms-24-11620-t001:** Demographic, inflammatory, functional, and clinical characteristics of obese and non-obese non-eosinophilic asthmatic patients, and obese and non-obese eosinophilic asthmatic patients (population 2; serum samples analysis).

	Obese Eosinophilic (n = 10)	Non-Obese Eosinophilic (n = 28)	Obese Non-Eosinophilic (n = 16)	Non-Obese Non-Eosinophilic (n = 23)	*p*-Value
Age (years) ^^^	50.4 (±13.8)	46.4 (±15.5)	53.9 (±8.4)	47 (±15.7)	N.S.
Female (%)	6 (60%)	20 (71.4%)	13 (81.3%)	15 (65.2%)	N.S.
BMI ^†^	34.5 (32.4–38.2)	23.6 (21.8–26.7)	32.2 (31–33.3)	23.8 (22.6–26)	****
Obesity (%)	10 (100%)	0 (0%)	16 (100%)	0 (0%)	****
Tobacco habit (%)					
Smokers	0 (0%)	3/27 (11.1%)	0/15 (0%)	0/20 (0%)	N.S.
Passive	0 (0%)	4/27 (14.8%)	2/15 (13.3%)	0/20 (0%)	N.S.
Ex-smokers	4 (40%)	8/27 (29.6%)	6/15 (40%)	7/20 (35%)	N.S.
Non-smokers	6 (60%)	12/27 (44.4%)	7/15 (46.7%)	13/20 (65%)	N.S.
Blood eosinophils (cells/µL) ^†^	600 (500–800)	700 (600–845)	150 (100–200)	100 (100–100)	****
Sputum eosinophils (%) ^†^	5% (2.5–20)	0% (0–48)	0% (0–6)	2% (0–5)	N.S.
Atopy (%)	7 (70%)	17 (60.7%)	13 (81.3%)	16 (69.6%)	N.S.
IgE (IU) ^†^	131 (55.6–347)	209 (87.4–753.5)	191 (61–670)	151 (53.3–358.3)	N.S.
FEV_1_/FVC (%) ^†^	85% (74.5–104)	78.3% (68–90)	88% (72–94.3)	80.3% (71–84.5)	N.S.
FeNO (ppb) ^†^	29 (21.5–41)	60 (34–82)	20.5 (11.1–30.8)	16 (12–23)	****
Exacerbations (%)	5 (50%)	17 (60.7%)	8 (50%)	9 (39.1%)	N.S.
Severity (%)					
Severe	3 (30%)	18/26 (69.2%)	5/15 (33.3%)	7/18 (38.9%)	*
Moderate	5 (50%)	5/26 (19.2%)	6/15 (40%)	4/18 (22.2%)	N.S.
Mild	2 (20%)	3/26 (11.5%)	4/15 (26.7%)	6/18 (33.3%)	N.S.
Intermittent	0 (0%)	0/26 (25%)	0/15 (0%)	1/18 (5.6%)	N.S.
ACT ^†^	20.5 (15–23)	21 (14.5–23.5)	21 (17–22)	22 (18–25)	N.S.
ICS and LABA (%)	10 (100%)	25 (89.3%)	15 (93.8%)	19 (82.6%)	N.S.
Late asthma onset	8/9 (88.9%)	20/25 (80%)	9/15 (60%)	14/18 (77.8%)	N.S.

Results are expressed as ^^^ mean (±SD) or ^†^ median (IQR); N.S., Non-significant; ****, *p* < 0.0001; *, *p* < 0.05; BMI, body mass index; FEV1, forced expiratory volume measured during the first second; FVC, forced vital capacity; FeNO, fractional exhaled nitric oxide; Ppb, parts per billion; ICS and LABA, inhaled corticosteroids and long-acting β2-agonists; and ACT, asthma control test.

**Table 2 ijms-24-11620-t002:** ROC curves for different comparisons between groups of asthmatic patients: obese and non-obese non-eosinophilic asthmatic patients, obese and non-obese eosinophilic asthmatic patients, and obese and non-obese in all asthmatic patients.

	Hsa-miR-26a-1-3p		Hsa-miR-376a-3p	
Comparison Groups	AUC	*p*-Value	AUC	*p*-Value
Obese non-eosinophilic vs. Non-obese non-eosinophilic	**1.00**	**<0.01**	0.50	>0.05
Obese eosinophilic vs. Non-obese eosinophilic	0.56	>0.05	0.63	>0.05
Non-obese non-eosinophilic vs. Non-obese eosinophilic	**0.91**	**<0.001**	**0.71**	**<0.05**
Obese non-eosinophilic vs. Obese eosinophilic	0.50	>0.05	0.64	>0.05
Obese non-eosinophilic vs. Non-obese eosinophilic	0.58	>0.05	**0.75**	**<0.01**
Non-obese non-eosinophilic vs. Obese eosinophilic	**1.00**	**<0.05**	0.61	>0.05
Obese vs. Non-obese	0.66	>0.05	0.60	>0.05

AUC, area under curve. Bold values are statistically significant (*p* < 0.05). Green color intensity indicates a higher positive correlation.

## Data Availability

The data that support the findings of this study are available from the corresponding author, V.d.P., upon reasonable request.

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
