# Peer review of "Obese Asthma Phenotype Is Associated with hsa-miR-26a-1-3p and hsa-miR-376a-3p Modulating the IGF Axis"

_ijms, 2023, doi:10.3390/ijms241411620_

Round 1

Reviewer 1 Report

The paper aimed to study the association between the expression of hsa-miR-26a-1-3p, hsa-miR-376a-3p and their target genes in asthmatic subjects with or without obesity in search for possible biomarkers. The paper is interesting, well-written and addresses important research question about the obesity-related asthma. However, there are some unclear fragments that should be explained or addressed.

Methods section:

- study design is a bit confusing, it is not clear if lung tissue and serum are from the same patients or do they comprise different populations? In table 1c 4 groups are described, whereas supplementary table 1a and 1b contains only 2 groups (asthmatic versus controls and obese vs non-obese)? A scheme that presents groups (with the exact number of patients per group) and which material (lung tissue or serum) was used from which patients at each stage of the study would be helpful

- as miRNA expression may be affected by drugs that patients received, the list of drugs should be mentioned, which drugs and at what doses for the patients in each group

- why equal volumes of serum for qPCR were used? The Authors should explain why miRNA concentration was not measured and used for equivalent amounts for serum as they did for tissue?

- how the internal control was optimized/verified as a good endogenous control gene? Could the Authors present the results from normalization analysis (e.g. from Genorm, Normfinder, Bestkeeper)

- the justification for target genes selection of studied miRNAs should be included: did the Author choose the genes that were previously validated experimentally as targets for these miRNAs? If not, what criteria and which tools did they use to select those genes as potential miRNAs targets?

Results:

- the significant differences in age between groups  may have significantly affected the expression of miRNA – did the authors use this variable as a confounder to assess the true effect of obesity (taking age into account) on miRNA expression? If not, this should be mentioned as a limitation of this study

- what about neutrophils? If blood morphology was done it would be useful also to report neutrophils in obese and non-obese patients (table S1b), as the previous studies indicated that obesity-related asthma may have neutrophilic inflammation component

- tab S1a – looking at the study aim the Authors should add BMI in the table for lung biopsy samples in asthmatic and control group to verify if this variable was accounted for; table S1b – it is surprising that non-obese patients had worse spirometry results – this result should be discussed in the Discussion section

Discussion:

- the Authors should explain why increased expression of miRNA-26a-1-3p resulted in increased expression of IGFBP3? MiRNAs regulate negatively gene expression i.e. by decreasing transcript stability or preventing translation, so increased expression of miRNA should decrease the expression of target gene/protein – this finding should be discussed in more detail

- conclusions – taking into account the findings i.e. the most significant differences in the expression of studied miRNAs were observed between eosinophilic and non-eosinophilic groups, whereas ROC curves for Obese non-eosinophilic vs. Obese eosinophilic were not significant. Therefore, conclusions should be formulated more cautiously

Minor issues:

- line 342 should be endogenous control (synonym for the reference gene) not exogenous

- it is not clear what was the reason to dilute the cDNA samples 1:30 or 1:60? This should be clarified.

No comments.

Author Response

Reviewer 1:

The paper aimed to study the association between the expression of hsa-miR-26a-1-3p, hsa-miR-376a-3p and their target genes in asthmatic subjects with or without obesity in search for possible biomarkers. The paper is interesting, well-written and addresses important research question about the obesity-related asthma. However, there are some unclear fragments that should be explained or addressed.

Dear reviewer, thank you very much for all your comments. Thanks to them the article will have a higher quality.

Major issues:

Methods section:

  1. Study design is a bit confusing, it is not clear if lung tissue and serum are from the same patients or do they comprise different populations? In table 1c 4 groups are described, whereas supplementary table 1a and 1b contains only 2 groups (asthmatic versus controls and obese vs non-obese)? A scheme that presents groups (with the exact number of patients per group) and which material (lung tissue or serum) was used from which patients at each stage of the study would be helpful.

Answer: I am sorry that the study design is somewhat confusing. I will try to clarify, as much as possible, what you are proposing to us. The lung tissue and serum samples are not from the same patients; the samples are from two different populations.

We have mentioned in the manuscript file (and in the supplementary tables file) to which population each table refers, in the results section, and in the materials and methods section (study subjects and sample collection). This can be seen on page 2, lines 82-83 and 91-92; on page 3, line 98; on page 9, line 362; and on page 10, line 373 (main document with changes or modifications). Table S1a refers to population 1, the patient population from which lung tissue samples were obtained. These lung tissue samples were obtained from two groups, asthmatic subjects, and individuals without a diagnosis of asthma (controls). Table S1b refers to population 2, the patient population from which serum samples were obtained. These serum samples were obtained from two groups, obese and non-obese asthma subjects. Table 1c (hereafter referred to as 1) refers to population 2, the patient population from which serum samples were obtained. In this case, these are the same individuals as in the previous case, but the obese and non-obese asthma subjects are further subdivided into eosinophilic and non-eosinophilic asthma individuals (hence 4 groups).

As suggested, a scheme is elaborated, to facilitate the understanding of the study design, with the groups (and the number of patients in each group) and the material (lung tissue samples or serum samples) that was used at each stage of the study. This scheme is proposed as a complementary figure (Figure S1), as indicated on page 10, lines 381-383 (document with changes or modifications). See in the attached document.

  1. As miRNA expression may be affected by drugs that patients received, the list of drugs should be mentioned, which drugs and at what doses for the patients in each group.

Answer: As you comment, the expression of miRNAs can be affected by patient treatment. Therefore, Tables S1b and 1c (Table 1 hereafter), which refer to patients from whom serum samples were obtained (hereafter referred to as population 2), show the number of subjects, in each of the groups, receiving Inhaled Corticosteroids (ICS) and Long-Acting β2-Agonists (LABA). Since no significant differences are observed in relation to patient treatment, it could be considered that the changes in the expression of miRNAs are not due to treatment. This is mentioned in the discussion, on page 7, lines 252-254. The dose is not shown since, unfortunately, we do not have this data.

In the case of the patients from whom lung tissue samples were obtained, regrettably we do not have information on the patients' treatment.

  1. Why equal volumes of serum for qPCR were used? The Authors should explain why miRNA concentration was not measured and used for equivalent amounts for serum as they did for tissue?

Answer: In response to your question, qPCR, both in serum and lung tissue samples, was performed according to the instructions of the manufacturer of the miRCURY LNA SYBR Green PCR Kit (Qiagen).

Since the concentrations of miRNAs in serum are very low, making their measurement unreliable, the same volume of each sample is used to normalise the amount of miRNAs across all samples, as well as others1:

  1. Kroh EM, Parkin RK, Mitchell PS, Tewari M. Analysis of circulating microRNA biomarkers in plasma and serum using quantitative reverse transcription-PCR (qRT-PCR) [published correction appears in Methods. 2010 Nov;52(3):268]. Methods. 2010;50(4):298-301. doi:10.1016/j.ymeth.2010.01.032.

Moreover, to ensure that differences in concentration, due to choosing the same volume in the case of serum samples, do not affect the miRNA expression measurements, it is normalised with endogenous, which function as control for the qPCR reaction. This is commented on page 10, lines 404-407 (document with changes or modifications).

  1. How the internal control was optimized/verified as a good endogenous control gene? Could the Authors present the results from normalization analysis (e.g. from Genorm, Normfinder, Bestkeeper).

Answer: In reply to your question, to select endogenous and/or exogenous genes as internal controls for the reaction, the statistical software GraphPad Prism® v6-8.0 (GraphPad Software Inc., San Diego, CA) was used. A homogeneity of means test is performed on the Ct of the total samples from each of the two comparison groups and, if the difference in means is not statistically significant, it is assumed to be a stable endogenous and/or exogenous that can be used as an internal control for the reaction, as others have done1. In the qPCR of the serum samples, the combination of the miRNAs hsa-miR-103a-3p and hsa-miR-191-5p were used as explained above, while for the qPCR of the lung tissue samples, in addition to these two, U6 snRNA and cel-miR-39-3p were used.

  1. Mestdagh P, Van Vlierberghe P, De Weer A, Muth D, Westermann F, Speleman F, Vandesompele J. A novel and universal method for microRNA RT-qPCR data normalization. Genome Biol. 2009;10(6):R64. doi: 10.1186/gb-2009-10-6-r64

Nevertheless, as you indicate, I present below the results from normalisation analysis, using the Bestkeeper tool. This is referred to in the text on page 10, lines 410-413 (document with changes or modifications).

  • Analysis population 1 (lung biopsy samples)

hsa-miR-103a-3p

hsa-miR-191-5p

U6 snRNA

cel-miR-39-3p

Average of all

HKG 1

HKG 2

HKG 3

HKG 4

HKG 5

n

36

36

35

36

36

geo Mean [CP]

22.14

23.13

29.75

27.66

25.64

ar Mean [CP]

22.15

23.14

29.79

27.66

25.65

min [CP]

19.55

20.64

26.03

25.91

23.03

max [CP]

23.41

24.04

34.60

27.97

27.03

std dev [± CP]

0.49

0.44

1.36

0.20

0.50

CV [% CP]

2.20

1.90

4.56

0.73

1.94

vs.

HKG 1

HKG 2

HKG 3

HKG 4

HKG 4

HKG 2

0.853

-

-

-

-

p-value

0.001

-

-

-

-

HKG 3

0.448

0.368

-

-

-

p-value

0.007

0.030

-

-

-

HKG 4

0.659

0.646

0.382

-

-

p-value

0.001

0.001

0.024

-

-

HKG 5

0.757

0.683

-17.475

0.666

-

p-value

0.001

0.000

#¡NUM!

0.001

-

BestKeeper vs.

HKG 1

HKG 2

HKG 3

HKG 4

HKG 5

coeff. of corr. [r]

0.794

0.724

0.825

0.685

0.998

p-value

0.001

0.001

0.001

0.001

0.001

TG 1

TG 2

TG 3

TG 4

TG 5

vs.

vs.

vs.

vs.

vs.

BK

BK

BK

BK

BK

coeff. of corr. [r]

0.794

0.724

0.825

0.685

0.998

coeff. of det. [r^2]

0.630

0.524

0.681

0.469

0.996

intercept [CP]

1.514

4.934

-23.044

18.361

-0.598

slope [CP]

0.809

0.714

2.071

0.365

1.030

SE [CP]

±0.424

±0.465

±0.965

±0.265

±0.045

p-value

0.001

0.001

0.001

0.001

0.001

Power [x-fold]

1.75

1.64

4.20

1.29

2.04

Based on the mentioned above, regarding the analysis of means, and what was observed in the Bestkeeper results, it seems appropriate to choose the mean of the 4 RNAs as internal control of the reaction since it does not present a high standard deviation, and the correlation coefficient is fine.

  • Analysis population 2 (serum samples)

hsa-miR-103a-3p

hsa-miR-191-5p

Average of both

HKG 1

HKG 2

HKG 3

n

77

77

77

geo Mean [CP]

23.59

24.24

23.92

ar Mean [CP]

23.63

24.28

23.96

min [CP]

20.86

21.39

21.13

max [CP]

26.99

28.00

27.47

std dev [± CP]

1.21

1.24

1.20

CV [% CP]

2.20

1.90

4.56

vs.

HKG 1

HKG 2

HKG 3

HKG 2

0.949

-

-

p-value

0.001

-

-

HKG 3

0.987

0.988

-

p-value

0.001

0.001

-

BestKeeper vs.

HKG 1

HKG 2

HKG 3

coeff. of corr. [r]

0.987

0.987

1.000

p-value

0.001

0.001

#¡DIV/0!

TG 1

TG 2

TG 3

vs.

vs.

vs.

BK

BK

BK

coeff. of corr. [r]

0.987

0.987

1.000

coeff. of det. [r^2]

0.974

0.974

1.000

intercept [CP]

-0.024

0.023

-0.006

slope [CP]

0.988

1.013

1.000

SE [CP]

±0.239

±0.245

±0

p-value

0.001

0.001

#¡DIV/0!

Power [x-fold]

1.98

2.02

2.00

Likewise, based on the previous mean analysis and what was observed in the results of Bestkeeper, it appears most adequate to use the 2 RNAs mean as internal control of the reaction because it presents the lowest standard deviation, and the correlation coefficient is proper.

  1. The justification for target genes selection of studied miRNAs should be included: did the Author choose the genes that were previously validated experimentally as targets for these miRNAs? If not, what criteria and which tools did they use to select those genes as potential miRNAs targets?

Answer: In response to your question, the online bioinformatics tool DIANA-miRPath v3.0 was used to select genes as potential miRNA targets. This informatics resource yields the signaling pathways, altered by deregulated miRNAs, and the genes involved in them. Pathways relevant to asthma pathology were represented when a p-value <0.05 was seen. Furthermore, Zehavi et al. identified the IGF1R gene as a target of hsa-miR-376a and hsa-miR-376c miRNAs. Since the protein encoded by the IGF1R gene is involved in the IGF axis, it was chosen, together with IGFBP-3, for ELISA evaluation.

The justification, discussed here, for the selection of miRNA target genes and proteins (tools and criteria for selection) has been added and can be found on page 11, lines 432-433, 435-436 and 438-440 (document with changes or modifications).

Results:

  1. The significant differences in age between groups may have significantly affected the expression of miRNA – did the authors use this variable as a confounder to assess the true effect of obesity (taking age into account) on miRNA expression? If not, this should be mentioned as a limitation of this study.

Answer: As you suggest, we fully agree that this is mentioned as a limitation of this study. While it is true that there are significant differences in age between the asthmatic and control groups, from which biopsies were obtained, in the obese and non-obese groups (with larger numbers of subjects) from which serum samples were employed, or in the 4 groups, once subdivided, no significant differences in age are observed and, because of this, age may not significantly affect miRNA expression. This is reflected in the limitations of the study, as can be seen on page 9, lines 350-357 (document with changes or modifications).

  1. What about neutrophils? If blood morphology was done it would be useful also to report neutrophils in obese and non-obese patients (table S1b), as the previous studies indicated that obesity-related asthma may have neutrophilic inflammation component.

Answer: As you say, it would be useful to report the blood neutrophil counts of obese and non-obese asthma patients (table S1b) because, as you point out, obesity-associated asthma may have a neutrophilic inflammatory component. We have not been able to include the number of neutrophils in the blood since, unfortunately, we do not have this data available.

However, although the number of neutrophils in blood is not available, according to the percentage of neutrophils in sputum, in our study the 27 % of patients with obese asthma are individuals with a neutrophilic component. This was mentioned in the discussion, on page 8, lines 265-267 (document with changes or modifications).

In addition, in the limitations section, we have added that, unfortunately, we do not have all the clinical data available. This is reflected in the limitations of the study, as can be seen on page 9, lines 357-358 (document with changes or modifications).

  1. Tab S1a – looking at the study aim the Authors should add BMI in the table for lung biopsy samples in asthmatic and control group to verify if this variable was accounted for; table S1b – it is surprising that non-obese patients had worse spirometry results – this result should be discussed in the Discussion section.

Answer: Considering the aim of the study, as you state, it would be interesting to add BMI in table S1a but, regrettably, we do not have this information and have not been able to add it. On the other hand, indeed, when comparing obese and non-obese asthma patients, a worse lung function outcome is seen in non-obese asthma patients, which is surprising. This may be due, to some extent, to the fact that there are more non-obese than obese asthma patients and, within the group of non-obese asthma patients, a high number of patients with eosinophilic asthma (that is considered a severe asthma phenotype), which may cause these non-obese asthma patients to have worse lung function since, when subdividing obese and non-obese asthma patients into eosinophilic and non-eosinophilic, the eosinophilic asthma patient groups show worse lung function than the non-eosinophilic asthma patient groups. As you suggested, we comment this in the discussion, which can be found on page 7, lines 254-263; and on page 8, lines 264-265 (document with changes or modifications).

Discussion:

  1. The Authors should explain why increased expression of miRNA-26a-1-3p resulted in increased expression of IGFBP3? MiRNAs regulate negatively gene expression i.e. by decreasing transcript stability or preventing translation, so increased expression of miRNA should decrease the expression of target gene/protein – this finding should be discussed in more detail.

Answer: Given that, as you note, miRNAs generally behave as silencers of their target mRNAs, an inverse correlation between miRNA-target gene would be expected. However, miRNAs-mRNAs relationships are complex, and the lung tissue samples (in this case) in which they have been studied, miRNA and target gene, differ from each other, and there may be other miRNAs involved in the modulation of a particular mRNA (IGFBP-3 mRNA, in this case) and also other target genes of this miRNA (hsa-miR-26a-1-3p, in this case) that could be inhibitors of the target gene we are talking about, IGFBP3. In this way, we could explain the positive correlation between hsa-miR-26a-1-3p and its target gene, IGFBP3, an explanation that is found on page 4, lines 129-137 (document with changes or modifications).

Conclusions:

  1. Taking into account the findings i.e., the most significant differences in the expression of studied miRNAs were observed between eosinophilic and non-eosinophilic groups, whereas ROC curves for Obese non-eosinophilic vs. Obese eosinophilic were not significant. Therefore, conclusions should be formulated more cautiously.

Answer: As you comment, the most significant differences in the expression of the two miRNAs studied were observed between the groups of patients with eosinophilic and non-eosinophilic asthma, whereas not all ROC curves were significant. Therefore, we have reformulated the conclusion to be more specific and in accordance with our results rather than generic. This can be seen on page 12, lines 476-480 (document with changes or modifications).

Minor issues:

  1. Line 342 should be endogenous control (synonym for the reference gene) not exogenous.

Answer: According to this comment, the term exogenous is used since synthetic Uni-Sp6 and miRNA cel-miR-39-3p (Qiagen) were added to the reaction mixture prior to the reverse transcription to verify that the reverse transcription was performed correctly after the qPCR has been completed. However, we have removed the term exogenous in case it causes confusion. This can be seen on page 10, line 399 (main document with changes or modifications).

  1. It is not clear what was the reason to dilute the cDNA samples 1:30 or 1:60? This should be clarified.

Answer: In answer to your question, cDNA from serum or lung tissue samples is diluted 1:30 or 1:60, respectively, according to the miRCURY LNA SYBR Green PCR Kit (Qiagen) producer's recommendations. We have added this in the manuscript, which can be found on page 10, lines 403-404 (document with changes or modifications).

Reviewer 2 Report

General comments:

This manuscript focuses on a couple of microRNAs (miRNAs) and their associations with asthma, particularly obesity in asthma. Hsa-miR-26a-1-3p and -376a-30, which the authors have studied before, were more highly expressed in asthmatic than healthy lungs. One of the target genes, IGFBP3, had higher gene expression in asthma compared to healthy but IGFBP3 protein concentration was lower in serum from obese patients with asthma than non-obese ones, and correlated inversely with BMI, whereas IGF1R was higher. ROC analyses were performed and predictive ability of the miRNAs for, e.g., obesity, in eosinophilic and non-eosinophilic asthma are shown. It is suggested that these two miRNAs could be used as biomarkers to phenotype patients with asthma in relation to obesity. However, one is left with a somewhat unclear impression, partly that these miRNAs and their targets in the IGF axis are associated with asthma and phenotypes in some manner, although partly that the association(s) is/are not very straightforward.

Specific comments:

Major comments:

11)     Figure 1: Please display these data differently, preferably so one can see the individual data points and variation in both the asthma and non-asthma groups.

22)      Fig. 3: Please restrict the x axes, while still encompassing all data, so the distribution of the data points are easier to see, for instance, from a BMI of about 15 to 40 or so. It appears from the text that b shows only non-eosinophilic asthma, although that’s currently not stated in the legend. For completion, please also make a table showing correlations for all asthma, eosinophilic asthma, and non-eosinophilic asthma.

33)      Table 2: Please also include data for obese vs non-obese in all asthma.

44)      Results, section 2.2, lines 109-111: Please provide more information on the method and results of this in silico analysis, including placing text in the Methods section.

55)      Title: The wording, especially “governed by” sounds like an exaggeration. A modified, toned down title would be preferable. Perhaps “is associated with” instead of “is governed by”.

Minor comments:

66)      Fig. 2: a-b and c-d seem to be switched in the legend compared to the actual figure. Please correct.

77)      Introduction, lines 53-57: Please provide one or more literature references for this sentence on systemic inflammation including IL1beta, IL6, and more in asthma.

88)      Similarly, in the Discussion, although the IGF axis etc. is already discussed, it would be helpful if the authors expanded a bit more on what is known about the IGF axis, insulin and insulin resistance particularly in asthma, including severe asthma, and also how their results fit into this context.

99)      Table S1b: Please comment on the differences in lung function and severity between obese and non-obese? Did this come about by chance or were obese patients deliberately recruited to be less severe than the non-obese?

110)  “Table 1 c”: There doesn’t seem to be any reason to call this table “1c”. Why not just Table 1?

Author Response

Reviewer 2:

General comments:

This manuscript focuses on a couple of microRNAs (miRNAs) and their associations with asthma, particularly obesity in asthma. Hsa-miR-26a-1-3p and -376a-30, which the authors have studied before, were more highly expressed in asthmatic than healthy lungs. One of the target genes, IGFBP3, had higher gene expression in asthma compared to healthy but IGFBP3 protein concentration was lower in serum from obese patients with asthma than non-obese ones, and correlated inversely with BMI, whereas IGF1R was higher. ROC analyses were performed and predictive ability of the miRNAs for, e.g., obesity, in eosinophilic and non-eosinophilic asthma are shown. It is suggested that these two miRNAs could be used as biomarkers to phenotype patients with asthma in relation to obesity. However, one is left with a somewhat unclear impression, partly that these miRNAs and their targets in the IGF axis are associated with asthma and phenotypes in some manner, although partly that the association(s) is/are not very straightforward.

Dear reviewer, thank you very much for all your comments. Thanks to them the article will have a higher quality.

Specific comments:

Major comments:

  1. Figure 1: Please display these data differently, preferably so one can see the individual data points and variation in both the asthma and non-asthma groups.

Answer: We have modified the figure. As you recommended, we have displayed the data so that one can see the individual data points and variation in both the asthma and non-asthma (control) groups. This can be seen on page 3, line 106; on page 4, lines 124, 138, 140 and 142-143; and in figure 1 (main document with changes or modifications). See attached file.

  1. Fig. 3: Please restrict the x axes, while still encompassing all data, so the distribution of the data points are easier to see, for instance, from a BMI of about 15 to 40 or so. It appears from the text that b shows only non-eosinophilic asthma, although that’s currently not stated in the legend. For completion, please also make a table showing correlations for all asthma, eosinophilic asthma, and non-eosinophilic asthma.

Answer: In accordance with this comment, we have constrained the x-axis so that it encompasses all data points, making it easier to visualize. We have also modified the figure legend, adding which population is involved in each case (3a and 3b). As you told me, below the graphs we have added a table (3c) with the correlation data of IGFBP-3 concentration and hsa-miR-26a-1-3p expression levels (2-ΔCt) with BMI in the two groups separately and in the total population. In red are shown those correlations that are negative and in green those that are positive. In addition, bold values are statistically significant (p < 0.05). This can be seen on page 5, lines 183 and 185; on page 6, lines 186-192; and in figure 3 (main document with changes or modifications). See attached file.

  1. Table 2: Please also include data for obese vs non-obese in all asthma.

Answer: Following this comment, we have included in the table the data of the ROC curves for the comparison of all obese and non-obese patients. This can be seen on page 6, lines 195-197 and 207-208; and in table 2 (main document with changes or modifications).

Hsa-miR-26a-1-3p

Hsa-miR-376a-3p

Comparison groups

AUC

P-value

AUC

P-value

Obese non-eosinophilic vs. Non-obese non-eosinophilic

1.00

< 0.01

0.50

> 0.05

Obese eosinophilic vs. Non-obese eosinophilic

0.56

> 0.05

0.63

> 0.05

Non-obese non-eosinophilic vs. Non-obese eosinophilic

0.91

< 0.001

0.71

< 0.05

Obese non-eosinophilic vs. Obese eosinophilic

0.50

> 0.05

0.64

> 0.05

Obese non-eosinophilic vs. Non-obese eosinophilic

0.58

> 0.05

0.75

< 0.01

Non-obese non-eosinophilic vs. Obese eosinophilic

1.00

< 0.05

0.61

> 0.05

Obese vs. Non-obese

0.66

> 0.05

0.60

> 0.05

  1. Results, section 2.2, lines 109-111: Please provide more information on the method and results of this in silico analysis, including placing text in the Methods section.

Answer: Considering what you tell us, we have added all the results provided by the online tool DIANA-miRPath v3.0. We have included, in the results section, all the signalling pathways regulated by these two miRNAs, indicating those linked to asthma pathology and therefore of interest, and the genes involved in these pathways. In addition, we have included further information in the materials and methods section more information from the in silico analysis. This can be seen on page 3, lines 112-122; and on page 11, lines 432-433, 435-436 and 438-440 (main document with changes or modifications).

  1. Title: The wording, especially “governed by” sounds like an exaggeration. A modified, toned down title would be preferable. Perhaps “is associated with” instead of “is governed by”.

Answer: As you suggest, to tone down the title, we have changed the "is governed by" to "is associated with". This can be seen on page 1, line 2 (main document with changes or modifications).

Minor comments:

  1. Fig. 2: a-b and c-d seem to be switched in the legend compared to the actual figure. Please correct.

Answer: As you state, a-b and c-d are switched in the legend of the figure compared to the actual figure. We have modified them. This can be seen on page 5, lines 161-163 (main document with changes or modifications).

  1. Introduction, lines 53-57: Please provide one or more literature references for this sentence on systemic inflammation including IL1beta, IL6, and more in asthma.

Answer: As you point out, a new literature reference to inflammatory components has been added. This can be seen on page 2, line 60; and on page 13, lines 532-533 (main document with changes or modifications).

Reference:

Bantulà M, Roca-Ferrer J, Arismendi E, Picado C. Asthma and Obesity: Two Diseases on the Rise and Bridged by Inflammation. J Clin Med. 2021;10(2):169. doi:10.3390/jcm10020169

  1. Similarly, in the Discussion, although the IGF axis etc. is already discussed, it would be helpful if the authors expanded a bit more on what is known about the IGF axis, insulin and insulin resistance particularly in asthma, including severe asthma, and also how their results fit into this context.

Answer: In relation to your point, information has been added on insulin, insulin resistance, IGF and asthma but, since there were not enough patients among the obese patients in the study to make statistics and we do not have data on these factors, it is not possible for us to discuss this information. This can be seen on page 8, lines 302-310 (main document with changes or modifications).

  1. Table S1b: Please comment on the differences in lung function and severity between obese and non-obese? Did this come about by chance or were obese patients deliberately recruited to be less severe than the non-obese?

Answer: In answer to your question, it is a chance finding as the choice of obese and non-obese individuals was random, based only on whether they were obese or not according to BMI.

On the other hand, in the discussion, reference is made to severity and lung function, as seen on page 7, lines 246-249 and 252-263; and on page 8, lines 264-265 (main document with changes or modifications).

  1. “Table 1 c”: There doesn’t seem to be any reason to call this table “1c”. Why not just Table 1?

Answer: As you say, there is indeed no reason to call table 1c as 1c. we have changed its name to, as you indicated, table 1. This can be seen on page 2, line 95; and on page 3, line 96 (main document with changes or modifications).

Round 2

Reviewer 2 Report

The authors have addressed the previous comments.